# Concurrence of stunting and overweight/ obesity among children: Evidence from Ethiopia

**Alinoor Mohamed Farah**[1]*, **Tahir Yousuf Nour**[1], **Bilal Shikur Endris**[2], **Seifu Hagos Gebreyesus**[2]

**1** Department of Public Health, College of Medicine and Health Sciences, Jigjiga University, Jigjiga, Ethiopia,
**2** Department of Reproductive Health and Health Service Management, School of Public Health, College of Health Sciences, Addis Ababa University, Addis Ababa, Ethiopia

* alinuriana@yahoo.com

**Data Availability Statement:** The data used in the study belong to a third party, The DHS Program, and is publicly available on http://dhsprogram.com/data/dataset/Ethiopia_Standard-DHS_2016.cfm.

## Abstract

### Background

Nutrition transition in many low- and middle-income countries (LMICs) has led to shift in childhood nutritional outcomes from a predominance of undernutrition to a double burden of under- and overnutrition. Yet, policies that address undernutrition often times do not include overnutrition nor do policies on overweight, obesity reflect the challenges of undernutrition. It is therefore crucial to assess the prevalence and determinants of concurrence stunting and overweight/obesity to better inform nutrition programs in Ethiopia and beyond.

### Methods

We analyzed anthropometric, sociodemographic and dietary data of children under five years of age from 2016 Ethiopian Demographic and Health Survey (EDHS). A total of 8,714 children were included in the current study. Concurrence of stunting and overweight/obesity (CSO) prevalence was estimated by basic, underlying and immediate factors. To identify factors associated with CSO, we conducted hierarchical logistic regression analyses.

### Results

The overall prevalence of CSO was 1.99% (95% CI, 1.57–2.53). The odds of CSO was significantly higher in children in agrarian region compared to their counter parts in the pastoralist region (AOR = 1.51). Other significant factors included; not having improved toilet facility (AOR = 1.94), being younger than 12 months (AOR = 4.22), not having history of infection (AOR = 1.83) and not having taken deworming tablet within the previous six months (AOR = 1.49).

### Conclusion

Our study provided evidence on the co-existence of stunting and overweight/obesity among infants and young children in Ethiopia. Therefore, identifying children at risk of growth flattering and excess weight gain provides nutrition policies and programs in Ethiopia and beyond

The DHS program is fully authorized to distribute the data, at no cost, upon registration on the program website https://dhsprogram.com/data/new-user-registration.cfm. Once an account has been created, users can request access to the Ethiopia DHS 2016 dataset by indicating their research need. Once the request has been received, it will be reviewed by The DHS Program staff within 1-2 business days and access will be granted if sufficient detail is provided. Those interested can access the data in the same manner as the authors. The authors had no special access privileges to the data that others would not have.

**Funding:** The author(s) received no specific funding for this work.

**Competing interests:** The authors have declared that no competing interests exist.

**Abbreviations:** AOR, Adjusted Odds Ratio; BAZ, Body Mass Index for Age Z-score; BMI, Body Mass Index; CI, Confidence Interval; COR, Crude Odds Ration; CSO, Concurrent of Stunting and Overweight/Obesity; EA, Enumeration Area; EDHS, Ethiopia Demographic and Health Survey; LAZ, Length for Age Z-score; HAZ, Height for Age Z-score; IRB, Institutional Review Board; LAC, Latin American and Caribbean; LMIC, Lower, Middle Income Countries; MENA, Middle-East and North Africa; NRERC, National Research Ethics Review Committee; UNICEF, United Nations Children's Fund; USAID, United States of Agency for International Development; WAZ, Weight for Age Z-score; WHZ, Weight for Height Z-score; WHO, World Health Organization.

with an opportunity of earlier interventions through improving sanitation, dietary quality by targeting children under five years of age and those living in Agrarian regions of Ethiopia.

## Introduction

Child malnutrition which includes both undernutrition and overweight are global challenges which is associated with an increased risk of mortality and morbidity, unhealthy development, and loss of productivity [1]. In 2016, nearly 155 million children under five years of age were reported stunted, while 41 million were obese or overweight [2]. Stunting and obesity distinctively pose a significant challenge to the health system, child survival and poor academic performance and a lower quality of life experience of children [3, 4] and their concurrence represents a serious public health challenge [5]. Though the drivers of these two forms of malnutrition appear distinct, new evidence indicates that there are shared biological, environmental and socioeconomic factors that contribute to the risk or prevalence of both [5, 6].

Although improvement of child undernutrition in Ethiopia has been achieved, stunting remains an important problem in Ethiopia, with 38% of children under five years of age affected. However, economic growth and urbanization in countries like Ethiopia have given rise to a nutrition transition, where there is a shift from traditional diets to "western diets" (energy-dense diets)which has led to an increase in overweight and obesity [7]. In Ethiopia, in addition to high levels of undernutrition, considerable levels of overweight/obesity have been observed. A recent metanalysis has shown a pooled prevalence of overweight and obesity of children and adolescents in Ethiopia to be 11.3% [8] and others reports showed a prevalence of overweight/obesity among preschoolers to be 13.8% in Gondar [9] and 7.3% in Hawassa [10]. On the other hand, undernutrition decreased but has not been as rapid as the rise in overweight and obesity, leading to a double burden of overnutrition and undernutrition [11].

Co-existence of two different forms of malnutrition is known as the double burden of malnutrition and could occur at country, household, or individual level [5]. At household level at least one member is undernourished and at least one member is overweight whereas at individual level the double burden of malnutrition often manifests as stunting or micronutrient deficiencies cooccurring with overweight or obesity [11–13]. At individual level, history of stunting coupled with consumption of high dense energy foods and micronutrient deficiencies owing to shared underlying determinants or physiologic links may also result in clustering of nutrition problems such as concurrence of stunting and overweight particularly among children under five years of age [11].

In that regard, the definition on the indicators used to explore and define the double burden of malnutrition phenomenon particularly at individual level stills remains unclear [14]. In other words, there is no uniform definition of double burden indicators. Indicators of child malnutrition used by the reviewed literatures are combination of height for age-z score (HAZ) and micronutrient deficiency [15], BMI for age-z-score and HAZ [16], weight for height z-score(WHZ) and HAZ [11, 17] and weight for age z-score (WAZ) and HAZ [18]. These differences in measurement, make comparison among studies difficult.

Children who are concurrently stunted and overweight/obese can be at greater risk of unhealthy development than normal children. In other words they are at high risk of noncommunicable disease since they impose a high metabolic load on a depleted capacity for homoeostasis [19]. Further, concurrent under and over nutrition is a public health challenge in a sense there is a need to strike a delicate balance between reducing undernutrition and preventing over nutrition [13].

There are substantial studies on double burden of malnutrition and mainly focused on prevalence and trends of double burden at different levels [7, 11–13, 15–18, 20–41] but few studies investigated factors associated with the double burden at individual level and in particular among children under five years of age [16, 26, 30, 33, 36].

In East Africa, studies on cooccurrence of stunting and overweight/obesity are few [40, 41] focused on prevalence and did not explore factors associated with it particularly among children under five years of age. In Ethiopia, no studies have been found that examined double burden of malnutrition among children under five years of age. The previous studies in Ethiopia focused on inclusive measure of child undernutrition and failed to capture the severity of malnutrition for some children who suffer from more than one type of malnutrition. The aim of this study is to assess the prevalence of concurrence of stunting and overweight/obese among children under five years age in Ethiopia and associated factors.

## Study subjects and methods

### Study design and data source

We used data set from EDHS which is a national representative cross-sectional household survey that was conducted from September to December 2015. A multistage stratified two- stage cluster sampling procedure was used to select samples.

In the first stage, a total of 645 enumeration areas (EAs) were selected from the sampling frame, 202 urban and 443 rural. A single EA covers 181 households and the sampling frame used was the 2007 Population and Housing Census. In the second stage of selection, a fixed number of 28 households were selected per EA. In all the selected households, anthropometric measurements were collected from children under five years of age [42].

Children's Data set which was based on woman and household questionnaires was used and included children under five years of age with complete anthropometric measurements. Based on UNICEF conceptual framework of malnutrition causation and literatures, a list of the potential predictors of double burden of malnutrition was developed [3, 43]. Based on the UNICEF conceptual framework, the variables were categorized into three groups: basic, underlying, and immediate factors. Thus, we analyzed anthropometric data of children under five years of age from 2016 EDHS which is nationally representative survey.

The survey was implemented by Central Statistical Agency (CSA) at the request of Ethiopian Federal Ministry of Health (FMoH) and funded by United States Agency for International Development (USAID).

### Study subjects

The available sample for children under five years of age was 10, 641, of the 10,641 children, we excluded from the analysis children with missing household data (n = 1170), children not alive at time of data collection (n = 635) and flagged cases (n = 122). The final data set comprised 8,714 children under five years of age (Fig 1).

### Anthropometric measures and interpretation

The outcome variable is concurrence of stunting and overweight/obesity (CSO) with the same child. Stunting was defined as height-for-age Z-score (HAZ) below -2SD and overweight/overweight was defined as BMI-for-age Z-score (BAZ) above 2SD from the respective WHO 2006 growth standards reference median [44]. Weight were measured using SECA scales while length/height was measured using Shorr measuring boards. Children younger than 24months were measured for length while lying down, and older children, while standing [42]. Two

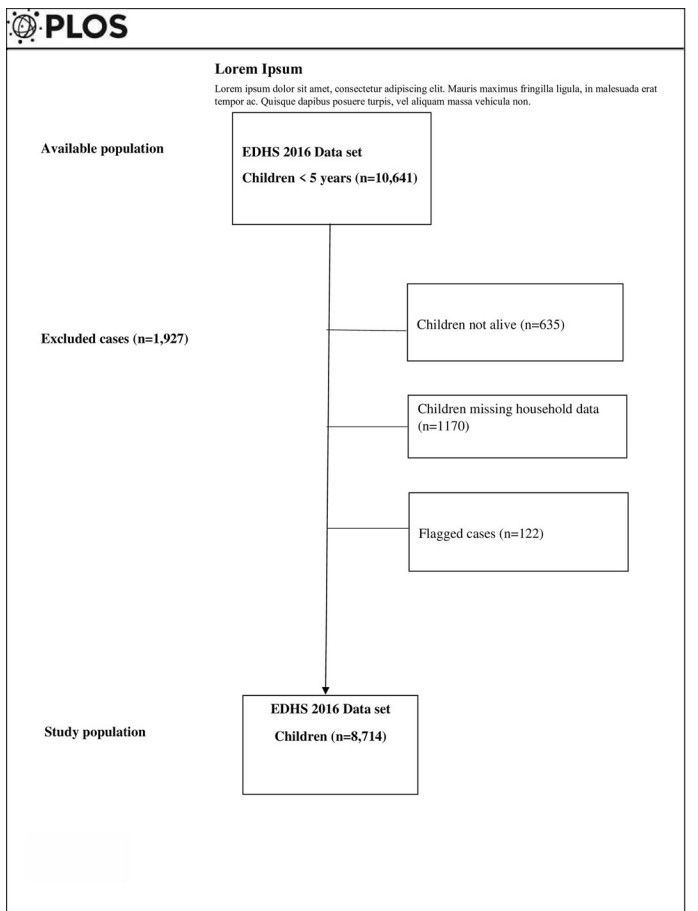

**Fig 1. Flow chart of sample selection.**

persons; one measurer and one recorder measure length/height: one to take measurements and other to record children's weight and length/height. Measurements were transformed into sex- and age-specific Z-scores using WHO 2006 growth standards [44].

## Assessment of associated factors

Maternal education, place of residence (urban and rural), household wealth, regions (agrarian and pastoralists), child's sex and age, latrine type, water source, birth size, history of infection and deworming, were factors included in the analysis. A detailed description these factors is presented in Table 1.

## Data collection procedure

The EDHS used standardized questionnaire to collect relevant demographic, health and nutrition data using trained data collectors who also take anthropometric measurement and blood sample.

The primary source of information on child related data was the caregiver (mostly the biological mother). When the biological mother was not present at the time of the data collection,

a family member who usually took care of the child was interviewed. The head of the house-hold usually the father or the mother were the primary source of information on household related data. All data were collected through house to house visit.

## Statistical analysis

Children recode data file in the form of STATA was used for analysis. Statistical analysis was performed using the STATA software package, version 14.1 (Stata Corp., College Station, TX, USA). Survey command (svy) was used to adjust for the complex sample design.

We estimated weighted prevalence of CSO by basic, underlying and immediate factors. Overall differences across the categories were statistically tested using design-based Pearson chi-squared test. Multiple hierarchical logistic regression was used to examine the effect of basic, underlying and immediate factors. First bivariate regression analyses were done for all potential predictors of stunting, overweight/obesity and CSO. CSO (Yes/No) was the depen-dent variable in each of the three regression models. Then, hierarchical regression models were run using variables which demonstrated P<0.20 during the bivariate regression analyses [45]. as levels such as 0.05 can fail in identifying variables known to be important [46, 47]. Regardless of significance, basic factors, as the primary independent variables, were retained in the final regression model. Age and sex being important factors for outcomes among children under five years of age, they were also retained in the final models.

The three-level hierarchical regression models were run following the recommendation of a previous study that suggested to take into account complex hierarchical relationships of differ-ent determinants at different level [48]. The first, second, and third models included basic, underlying, and immediate factors, respectively. Model-1, 2 and 3 included the basic, underly-ing and immediate factors which demonstrated P<0.20 during the bivariate regression analy-ses. To put it differently, we used model-1 to assess the overall effect of basic factors and excluded the underlying and immediate factors. We used model 2 to assess the effect of

**Table 1. Description of variables.**

| Sn | Variables | Description |
|---|---|---|
| 1 | Concurrent of stunting and overweight/obesity (CSO) | Defined when a child was both stunted and overweight/obese. |
|  | **Basic factors** |  |
| 2 | Residence | Categorized as urban and rural. |
| 3 | Household wealth category | Categorized as low, middle and high wealth categories |
| 4 | Caregiver education | Categorized as illiterate/none, primary, and secondary and above |
| 5 | Region | Categorized as mainly pastoral and mainly agrarian. Regions categorized under agrarian are Amhara, Tigray, SNNP, Benishangul, Gambela, Addis Ababa, Harari and Diredawa. Regions that classified under pastoralist are Somali and Afar. |
|  | **Underlying factors** |  |
| 6 | Water source | Categorized as improved and unimproved water source. |
| 7 | Toilet facility | Categorized as improved and unimproved toilet facility. |
| 8 | **Immediate factors** |  |
| 9 | Child sex | Categorized as boy and girl. |
| 10 | Child age | Categorized into <12,12–23, 24–35, 36–47 and 48–59 |
| 11 | Birth size | Categorized as large, average and small. |
| 12 | History of infection | Categorized as yes and no. |
| 13 | Deworming tablet use | Categorized as yes and no. |
| 14 | Iron supplement use | Categorized as yes and no. |

underlying factors in the presence of basic factors which were considered confounding factors and immediate factors were entered in model 3 in the presence of basic factors which were also considered confounding variables in model 3. Variable significant at p-value of 0.05 during the hierarchical regression analyses was considered to be determinant factor at each model in which the variable was first entered regardless of its performance in the subsequent model (s). For instance, if one of the factors in Model-1 is significant, its performance in the subsequent models will not matter. The approach was meant to avoid the possibility of underestimating the effects of basic factors [48].

### Ethical approval

Ethical clearance for the survey was provided by Institutional Review Board (IRB) of the College of Medicine and Health Sciences at Jigjiga University. Online application to analyze the secondary data was requested from DHS Program, USAID and we have been authorized to download data from the Demographic and Health Surveys (DHS) online archive.

## Result

### Background characteristics of children

The background characteristics of children and prevalence of CSO across different covariates are presented in Table 2. Fifty eight percent of children were male and the rest were female. Thirty one percent of children were younger than twelve months and the rest were older than twelve months. Majority of children lived in rural and agrarian regions.

### Prevalence of CSO

Overall, 38.3 and 3.86% of children were stunted and overweight respectively. The overall prevalence of CSO was 1.99% (95% CI,1.57–2.53). CSO prevalence among urban and rural children was 1.74 and 2.50%, respectively. Children in lowest wealth category and whom caregivers had no education had higher prevalence CSO compared to their counterparts. Children living in agrarian regions tend to suffer concurrently from stunting and overweight/obesity compared to their counterparts in pastoralist region. Children from households with no improved toilet facility also tend to have higher prevalence of CSO. Among boys and girls, the prevalence was 1.15 and 0.84% respectively. The age-specific estimates were 0.86 in those under 12 months of months. The prevalence of CSO by other child characteristics is shown in Table 2.

### Determinants of stunting

Table 3 shows determinants of stunting and obesity. The first, second, and third models contained basic, underlying, and immediate factors, respectively. Model-1 adjusted for basic factors only, Model-2 adjusted for basic factors and underlying factors and Model-3 adjusted for basic factors and immediate factors.

Determinants of stunting included lowest wealth index category (AOR = 1.57), children whom caregivers had no education (AOR = 3.14), not having improved toilet facility (AOR = 1.41), being a male child (AOR = 1.20), being at age group 24–35 (AOR = 6.03) and having been small at birth (AOR = 1.73).

### Determinants of overweight/obesity

Table 4 shows determinants of overweight/obesity. The first, second, and third models contained basic, underlying, and immediate factors, respectively. Model-1 adjusted for basic

**Table 2. Bivariate analysis of the relation of basic, underlying, and immediate factors with CSO.**

| | | Weighted frequency (%) | CSO prevalence (95% CI) | P-value* |
|---|---|---|---|---|
| *Basic factors (distal)* | | | | |
| Residence place | Urban | 12.7 | 2.50 (0.1, 4.5) | 0.64 |
| | Rural | 87.3 | 1.74(1.34, 2.26) | |
| Wealth | Low | 43.6 | 0.87(0.57, 1.33) | 0.24 |
| | Middle | 16.0 | 0.32 (0.2, 0.52) | |
| | High | 40.4 | 0.8(0.58, 1.12) | |
| Caregiver education | No | 62.5 | 1.25(0.93, 1.67) | 0.2 |
| | Primary | 33.6 | 0.67(0.45, 0.95) | |
| | Secondary+ | 3.9 | 0.08[0.02 0.28] | |
| Region | Agrarian | 97.1 | 1.93(1.51,2.47) | 0.02 |
| | Pastoral | 2.9 | 0.06(0.03,0.10) | |
| *Underlying factors (intermediate)* | | | | |
| Water source type | Not improved | 43.3 | 0.86(0.59, 1.26) | 0.91 |
| | Improved | 56.7 | 1.13(0.84, 1.52) | |
| Toilet facility source | Not improved | 93.8 | 1.87(1.45, 2.41) | 0.22 |
| | Improved | 6.2 | 0.12(0.06, 0.25) | |
| *Immediate factors (proximal)* | | | | |
| Child sex | Boy | 57.9 | 1.15(0.86, 1.54) | 0.18 |
| | Girl | 42.1 | 0.84(0.6, 1.18) | |
| Child age | <12 months | 31.1 | 0.62(0.43,0.88) | 0.01 |
| | 12–23 | 20.5 | 0.44(0.25,0.66) | |
| | 24–35 | 18.9 | 0.38(0.23,0.60) | |
| | 36–47 | 20.6 | 0.18(0.09,0.34) | |
| | 48–59 | 8.9 | | |
| Birth size | Large | 38.9 | 0.77(0.54, 1.11) | 0.1 |
| | Average | 43.7 | 0.87(0.63, 1.20) | |
| | Small | 17.4 | 0.35(0.22, 0.56) | |
| Infection history | No | 85.9 | 1.71(1.32,2.21) | 0.009 |
| | Yes | 14.1 | 0.28(0.17, 0.47) | |
| Deworming | No | 94.2 | 1.88(1.47, 2.40) | 0.122 |
| | Yes | 5.8 | 0.11(0.04, 0.30) | |
| Iron supplement | No | 90.1 | 1.79(1.38, 2.32) | 0.58 |
| | Yes | 9.9 | 0.2(0.1, 0.39) | |

CSO: Concurrent of stunting and overweight/obesity CI: confidence interval. *Based on Pearson chi-square test of association.

factors only, Model-2 adjusted for basic factors and underlying factors and Model-3 adjusted for basic factors and immediate factors.

Determinants of obesity included living in agrarian region (AOR = 1.51), being younger than 12 months (AOR = 3.74) and not having history of infection (AOR = 1.83).

## Determinants of CSO

The multiple hierarchical logistic regression model is presented in Table 5. The first, second, and third models contained basic, underlying, and immediate factors, respectively. Model-1 adjusted for basic factors only, Model-2 adjusted for basic factors and underlying factors and Model-3 adjusted for basic factors and immediate factors. The determinants of CSO included living in agrarian region (AOR = 1.51), not having improved toilet facility

**Table 3. Hierarchical multiple logistic regression analysis to determine basic, underlying and immediate determinants of stunting among children under five years of age.**

| Variables | | Model 1 | | Model 2 | | Model 3 | |
|---|---|---|---|---|---|---|---|
| | | COR (95% CI) | AOR (95% CI) | COR (95% CI) | AOR (95% CI) | COR (95% CI) | AOR (95% CI) |
| Residence | Urban | Ref | | | | | |
| | Rural | 2.06(1.82–2.34) | 1.12(0.95–1.34) | | | | |
| Wealth | Low | 1.89(1.70–2.09) | 1.57(1.37–1.79) ** | | | | |
| | Middle | 1.58(1.37–1.82) | 1.18(1 .00–1.38) * | | | | |
| | High | Ref | Ref | | | | |
| Caregiver education | No | 4.23(3.00–5.95) | 3.14(2.18–4.51) ** | | | | |
| | Primary | 3.26(2.29–4.62) | 2.52(1.75–3.62) ** | | | | |
| | Secondary+ | Ref | Ref | | | | |
| Region | Agrarian | 1.13(1.02–1.26) | 1.46(1.30–1.64) ** | | | | |
| | Pastoralist | | | | | | |
| Water source type | Not improved | | | 1.14(1.04–1.25) | 0.91(0.83–1.01) | | |
| | Improved | | | Ref | Ref | | |
| Toilet facility type | Not improved | | | 2.21(1.94–2.53) | 1.41(1.18–1.68) ** | | |
| | Improved | | | Ref | Ref | | |
| Child sex | Girl | | | | | Ref | |
| | Boy | | | | | 1.14(1.04–1.24) | 1.20(1.09–1.32) ** |
| Child age | <12 months | | | | | Ref | |
| | 12–23 | | | | | 3.68(3.12–4.33) | 3.76(3.16–4.49) ** |
| | 24–35 | | | | | 5.70(4.83–6.71) | 6.03(5.06–7.18) ** |
| | 36–47 | | | | | 5.13(4.36–6.05) | 5.43(4.56–6.47) ** |
| | 48–59 | | | | | 3.82(3.24–4.50) | 4.02(3.37–4.78) ** |
| Birth size | Large | | | | | Ref | Ref |
| | Average | | | | | 1.22(1.10–1.36) | 1.25(1.12–1.41) ** |
| | Small | | | | | 1.59(1.42–1.79) | 1.73(1.52–1.97) ** |
| Infection history | No | | | | | Ref | - |
| | Yes | | | | | 1.08(0.98–1.20) | - |
| Deworming | No | | | | | 0.94(0.83–1.08) | - |
| | Yes | | | | | Ref | - |
| Iron supplement | No | | | | | 0.93(0.79–1.10) | - |
| | Yes | | | | | Ref | - |

COR: Crude Odds Ratio; AOR: Adjusted odds ratio CI: Confidence interval *P-value significant when <0.05 ** P-value significant when <0.001.

Model-1: adjusted for residence place and wealth category.

Model-2: adjusted for residence place, wealth category and all variables under Model-2 .

Model-3: adjusted for residence place, wealth category and all variables under Model-3.

(AOR = 1.94), being younger than 12 months (AOR = 4.22), not having history of infection (AOR = 1.83) and not having taken deworming tablet within the previous six months (AOR = 1.49).

## Discussion

To our best of knowledge, this is a first study to show prevalence of CSO and its associated factors using a national representative sample of children under five years of age in Ethiopia. Our study provided evidence that there was concurrence of stunting and overweight among children under five years of age in Ethiopia. We also found CSO is associated with factors at different levels. The basic factor associated with higher odds of CSO was being

**Table 4. Hierarchical multiple logistic regression analysis to determine basic, underlying and immediate determinants of overweight/obesity among children under five years of age.**

| Variables | | Model 1 | | Model 2 | | Model 3 | |
|---|---|---|---|---|---|---|---|
| | | COR (95% CI) | AOR (95% CI) | COR (95% CI) | AOR (95% CI) | COR (95% CI) | AOR (95% CI) |
| Residence | Urban | 1.78(1.38–2.30) | 1.35(0.91–2.00) | | | | |
| | Rural | Ref | Ref | | | | |
| Wealth | Low | Ref | Ref | | | | |
| | Middle | 1.00(0.69–1.45) | 0.87(0.65–1.18) | | | | |
| | High | 1.63(1.27–2.07) | 1.44(0.64–2.05) | | | | |
| Caregiver education | No | Ref | Ref | | | | |
| | Primary | 1.04(0.79–1.39) | 0.77(0.51–1.16) | | | | |
| | Secondary+ | 1.91(1.14–3.19) | 1.26(0.91–1.75) | | | | |
| Region | Agrarian | 1.48(1.09–2.01) | 1.51(1.08–2.11) * | | | | |
| | Pastoralist | Ref | | | | | |
| Water source type | Not improved | | | 1.06(0.84–1.34) | - | | |
| | Improved | | | Ref | - | | |
| Toilet facility type | Not improved | | | Ref | | | |
| | Improved | | | 1.60(1.22–2.10) | 1.14(0.76–1.71) | | |
| Child sex | Girl | | | | | Ref | Ref |
| | Boy | | | | | 1.25(1.00–1.58) | 1.25(0.98–1.61) |
| Child age | <12 months | | | | | Ref | Ref |
| | 12–23 | | | | | 3.41(2.24–5.19) | 3.74(2.39–5.85) ** |
| | 24–35 | | | | | 2.62(1.70–4.05) | 2.67(1.67–4.25) ** |
| | 36–47 | | | | | 1.86(1.17–2.95) | 1.81(1.10–2.99) * |
| | 48–59 | | | | | 2.06(1.28–3.18) | 2.01(1.24–3.27) * |
| Birth size | Large | | | | | Ref | |
| | Average | | | | | 1.25(0.92–1.71) | - |
| | Small | | | | | 1.21(0.90–1.62) | - |
| Infection history | No | | | | | 1.72(1.28–2.33) | 1.83(1.33–2.53) ** |
| | Yes | | | | | Ref | Ref |
| Deworming | No | | | | | 1.58(1.04–2.39) | 1.49(0.93–2.39) |
| | Yes | | | | | Ref | Ref |
| Iron supplement | No | | | | | Ref | - |
| | Yes | | | | | 1.08(0.71–1.65) | - |

COR: Crude Odds Ratio; AOR: Adjusted odds ratio CI: Confidence interval *P-value significant when <0.05 ** P-value significant when <0.001.

Model-1: adjusted for residence place and wealth category.

Model-2: adjusted for residence place, wealth category and all variables under Model-2.

Model-3: adjusted for residence place, wealth category and all variables under Model-3.

from agrarian region. The underlying factor associated with CSO was not having improved toilet facility. The immediate factors found associated with higher odds of CSO were age younger than 12 months, no history of infection and not having received deworming for the last six months.

Our results showed high level of stunting among children under five years of age, with 38.3% of children being stunted. Overweight was also prevalent, though not as high as stunting. We found an overall prevalence of overweight/obesity was higher compared to estimates based on weight for height z-score. This finding is in agreement with a study that compared

**Table 5. Hierarchical multiple logistic regression analysis to determine basic, underlying and immediate determinants of CSO.**

| Variable | | Model 1 | | Model 2 | | Model 3 | |
|---|---|---|---|---|---|---|---|
| | | COR (95% CI) | AOR (95% CI) | COR (95% CI) | AOR (95% CI) | COR (95% CI) | AOR (95% CI) |
| Residence | Urban | 0.88(0.58–1.36) | 1.09(0.62–1.93) | | | | |
| | Rural | Ref | Ref | | | | |
| Wealth index | Low | Ref | Ref | | | | |
| | Middle | 1.11 (0.69–1.79) | 0.76(0.44–1.31) | | | | |
| | High | 1.10(0.77–1.58) | 1.07(0.68–1.68) | | | | |
| Caregiver education | No | 1.17(0.80–1.70) | 1.03(0.70–1.53) | | | | |
| | Primary | 0.95(0.39–2.36) | 0.74(0.27–1.99) | | | | |
| | Secondary + | Ref | Ref | | | | |
| Region | Agrarian | 1.57(1.02–2.42) | 1.51(1.08–2.11) * | | | | |
| | Pastoralist | Ref | Ref | | | | |
| Water source type | Not improved | | | 1.05(0.76–1.46) | - | | |
| | Improved | | | Ref | - | | |
| Toilet facility type | Not improved | | | 1.62(0.98–2.70) | 1.94(0.74–4.91) * | | |
| | Improved | | | Ref | Ref | | |
| Child sex | Girl | | | | | Ref | Ref |
| | Boy | | | | | 1.23(0.89–1.70) | 1.22(0.87–1.72) |
| Child age | <12 months | | | | | 4.47(2.37–8.42) | 4.22(2.22–8.05) ** |
| | 12–23 | | | | | 2.65(1.35–5.20) | 2.49(1.24–5.00) * |
| | 24–35 | | | | | 2.40(1.21–4.77) | 2.54(1.27–5.08) * |
| | 36–47 | | | | | 2.99(1.54–5.80) | 3.00(1.54–5.84) * |
| | 48–59 | | | | | Ref | Ref |
| Birth size | Large | | | | | Ref | Ref |
| | Average | | | | | 1.02(0.66–1.57) | - |
| | Small | | | | | 1.05(0.70–1.57) | - |
| Infection history | No | | | | | 1.88(1.20–2.94) | 1.98(1.25–3.15) * |
| | Yes | | | | | Ref | Ref |
| Deworming | No | | | | | 2.39(1.17–4.89) | 2.32(1.07–5.01) * |
| | Yes | | | | | Ref | Ref |
| Iron supplement | No | | | | | 0.80(0.45–1.42) | - |
| | Yes | | | | | Ref | - |

COR: Crude Odds Ratio; AOR: Adjusted odds ratio CI: Confidence interval *P-value significant when <0.05 ** P-value significant when <0.001.

Model-1: adjusted for residence place and wealth category.

Model-2: adjusted for residence place, wealth category and all variables under Model-2.

Model-3: adjusted for residence place, wealth category and all variables under Model-3.

changes in the prevalence of overweight in preschool children between 1990 and 2010 and found that estimates using BMI z-score were higher than those observed using WHZ [49].

The prevalence of children concurrently suffered stunting and overweight/obese was 1.99%. In comparison to other studies on prevalence of concurrence of stunting and overweight/obese among children under five years of age in African countries, the prevalence in our study was higher than studies conducted in Ghana and Kenya [16, 40] but lower than a study conducted in South Africa and Libya that reported prevalence of 18% and 7% respectively among children under five years of age [28, 38]. Similarly, when compared to studies from Asia and Latin America countries that determined the prevalence of concurrence of stunting and overweight/obese among children under five years of age, our study was lower

than a study conducted Ecuador that reported a prevalence of 2.8% [23], but higher than a study from Mexico that reported a prevalence of 1% [17] and lower than a prevalence from China that reported 5.06% [33]. A recent study that examined the double burden of malnutrition among children aged 6–59 months in the Middle-East and North Africa (MENA) and Latin American and Caribbean (LAC) regions also showed a prevalence that ranged from 0.4 to 10.7% in MENA regions and 0.3 to 1.9% in LAC regions [50]. The prevalence of our study is lower than prevalence of almost all LAC regions and within the range in most countries in MENA regions.

Our findings regarding factors associated with stunting, overweight/obesity and CSO are noteworthy. Children from agrarian region were more likely to be stunted, overweight/obese and concurrently stunted and overweight/obese. The finding was consistent with reports of previous studies in Ethiopia that showed higher prevalence of stunting in Amhara and Tigray, both Agrarian regions but lower stunting prevalence in Somali and Afar, both Pastoralist regions [42, 51]. This could be explained by the fact that short maternal stature is common in agrarian region [42, 51] than in pastoralists and thus intergenerational influence on height of their children since shorter parents are more likely to have shorter children [52]. Further, genetics and environmental factors are known to influence child's height [53, 54]. Another possible explanation could be due to difference in dietary practices. Pastoralists communities in Ethiopia are more likely to consume milk than agrarian communities. Animal source foods are rich in type I and particularly type II nutrients (the growth nutrients) nutrients and has shown to reduce risk of stunting in children [55, 56]. This is even evident in our sub-sample data; sub-sample analysis of data of current work for children aged 6–23 months using Pearson chi-squared test revealed that almost 62% of children from pastoralist communities consumed dairy products. Conversely, only 15% of children from agrarian communities consumed dairy products (P<0.001). Similarly, children residing in agrarian regions had higher odds of being overweight/obesity than their counter parts in pastoralist regions. This finding is similar to other studies conducted in Ethiopia that found higher prevalence of overweight/obesity among children residing in agrarian regions [9, 10]. This again could be explained by differences in lifestyle, diet or in feeding habits across regions. Studies conducted in agrarian regions found consumption of sweet food [9, 10] and early introduction of formula milk were among factors associated with overweight/obesity among preschoolers [10].

Children from poor households were more likely to be stunted than those from rich household. This could be due to the fact that infant and young child caring practices like hygiene, proper feeding and health services utilization are often poorly practiced among poor households compared to their richer counterparts [52, 57, 58]. This finding also agrees with the existing literature on stunting [16, 36]. Children from rural areas and whose mothers have not received formal education had also a higher risk of being stunted but not overweight/obese or CSO. Education and place residence are socio-economic indicators. Poverty and a lack of educational attainment is associated with poor nutrition and health practices such as poor nutrition across the life course due to inability to afford nutrient-rich foods [59].

Children from households with unimproved toilet facilities had higher odds of stunting and concurrently suffering from both stunting and overweight/obsess than children dwelling in households with improved toilet facilities. Access to improved sanitation services is crucial for preventing undernutrition since poor sanitation can result in childhood infection such as diarrhea which subsequently affect the linear growth of a child negatively [60, 61].

We also found higher odds of stunting in boys and those children above 12 months of age which is consistent with other studies [57, 62]. This could be due to the rampant suboptimal feeding practices in Ethiopia and high number of children are not meeting the minimum acceptable diet [42]. Conversely, higher odds of overweight and CSO was found in children

younger than 12 months of age which is in agreement with studies conducted in Ethiopia [63], Cameroon [64] and Malaysia [65]. After literature search, previous studies did not investigate association between overweight/obesity and different age groups and as why younger children are at higher risk of overweight/obese. Though one study related to increase of physical activity as the age increase which will lead to high metabolic activity and energy requirement [63]. Feeding pattern among these particular age group is one of the factors that might need to be explored to better understand such association.

Further, our finding of higher risks of stunting in boys than in girls was in agreement with previous reports which demonstrated higher odds of stunting in boys [52, 66]. This finding suggests that boys are generally vulnerable to malnutrition and could be a biological explanation. Epidemiological and cohort studies on neonatology demonstrated consistently higher morbidity and mortality in males than in early life [67–70] though the underlying mechanisms is poorly understood [70, 71].

Small birth size was also significantly associated with stunting. This finding was also in agreement with the existing evidence which suggests low birthweight linked to poor health and nutritional outcomes [1, 62, 72]. Fetal growth restriction is an important contributor to stunting in children and evidence showed that low birthweight was associated with 2.5–3.5-fold higher odds of wasting, stunting and underweight [72]. Further, other evidences have suggested low birth weight babies who exhibit catchup growth may be at risk of abnormal weight gain in childhood [73–77].

Deworming and not having history of infection were associated with CSO while not having history of infection was associated with overweight/obese. This could be possibly due to the fact that helminths and protozoans trigger leptin secretion which is related to inflammation, food intake and nutrient absorption and metabolism [78]. There is also evidence that all forms of chronic gut inflammation lead to growth faltering, whether indirectly by affecting nutrient balance or by more direct effects on metabolism [79].

The first 1,000 days is a critical time for physical and intellectual growth and set a foundation for long term heath and development [1]. As a result of greater awareness of significance of stunting as one of major public health problems, stunting reduction has gained increased international attention [80]. Equally, overweight/obesity in this age group deserves attention because at early stage of life catch up growth have been identified as one of the risk factors that lead to progression of abnormal weight gain [81–87]. Therefore, it is important to identify children that are at risk of developing CSO as early as possible to limit progression of both growth flattering and abnormal weight gain. Further, evidence have shown that if only under nutrition is targeted with an aim of targeting growth may unintentionally contribute to abnormal weight gain [88]. In this regard, humanitarian emergency nutrition programs that are currently focused mainly on food security can be used as platform to promote quality and nutritious diets and ensure that the food provided does not increase the risk of unhealthy diets which may unintentionally contribute to abnormal weight gains [5].

A recent analysis conducted by WHO has also showed that policies that address undernutrition often times do not include overweight/obesity and the vice versa [59]. Thus, integrated interventions, programs and policies that have the potential to improve the nutrition outcomes across the spectrum of malnutrition is of paramount importance to simultaneously reduce the risk or burden of both undernutrition and overweight/obesity [5].

Our study has the following limitations; first, recall bias while reporting the birth, infection, dietary history of children is still an issue of concern [89]. In other words, collecting data like birth size and history of infection are solely based on memory of mothers or the caretaker which might have led to recall bias. Further, due to the cross-sectional nature of this study a cause and effect relationship could not be inferred.

Given the above-mentioned limitations, the current work has some strengths. First, the EDHS data is a national representative data and conclusions about Ethiopia can be drawn. Second the data is reliable and of high quality since the standardized procedures are employed by such kind of survey. Third, appropriate statistical method was used to explore the relationships between the outcome variable and its determinants.

## Conclusion

In conclusion, our study provided evidence on the co-existence of stunting and overweight/obesity among children under five years of age in Ethiopia. CSO was associated with various factors originating from community and child levels. Therefore, identifying children at risk of growth flattering and excess weight gain provides nutrition policies and programs in Ethiopia and beyond with an opportunity of earlier interventions through improving sanitation, dietary quality by targeting children under five years of age and those living in Agrarian regions of Ethiopia.

## Acknowledgments

The authors will like to thank ICF international to grant permission to use the EDHS data. We will also like to thank Dr. Olusola Oladeji for editing the manuscript.

## Author Contributions

**Conceptualization:** Alinoor Mohamed Farah.

**Data curation:** Alinoor Mohamed Farah.

**Formal analysis:** Alinoor Mohamed Farah.

**Methodology:** Alinoor Mohamed Farah.

**Writing – original draft:** Alinoor Mohamed Farah.

**Writing – review & editing:** Alinoor Mohamed Farah, Tahir Yousuf Nour, Bilal Shikur Endris, Seifu Hagos Gebreyesus.

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
