## [Decision Letter · Decision Letter 0]

7 Apr 2020

PONE-D-20-03785

Concurrent of stunting and overweight/obesity among children:evidence from Ethiopia

PLOS ONE

Dear Mr Farah,

Thank you for submitting your manuscript to PLOS ONE. After careful consideration, we feel that it has merit but does not fully meet PLOS ONE’s publication criteria as it currently stands. Therefore, we invite you to submit a revised version of the manuscript that addresses the points raised during the review process.

We would appreciate receiving your revised manuscript by May 22 2020 11:59PM. To enhance the reproducibility of your results, we recommend that if applicable you deposit your laboratory protocols in protocols.io, where a protocol can be assigned its own identifier (DOI) such that it can be cited independently in the future. For instructions see: http://journals.plos.org/plosone/s/submission-guidelines#loc-laboratory-protocols

We look forward to receiving your revised manuscript.

Kind regards,

Nili Steinberg

Academic Editor

PLOS ONE

Additional Editor Comments (if provided):

Please see reviewer's comments

3. Your ethics statement must appear in the Methods section of your manuscript. If your ethics statement is written in any section besides the Methods, please move it to the Methods section and delete it from any other section. Please also ensure that your ethics statement is included in your manuscript, as the ethics section of your online submission will not be published alongside your manuscript.

Reviewers' comments:

Reviewer's Responses to Questions

**Comments to the Author**

1. Is the manuscript technically sound, and do the data support the conclusions?

Reviewer #1: No

Reviewer #2: Yes

Reviewer #3: Yes

2. Has the statistical analysis been performed appropriately and rigorously? 

Reviewer #1: I Don't Know

Reviewer #2: Yes

Reviewer #3: Yes

3. Have the authors made all data underlying the findings in their manuscript fully available?

Reviewer #1: No

Reviewer #2: Yes

Reviewer #3: Yes

4. Is the manuscript presented in an intelligible fashion and written in standard English?

Reviewer #1: No

Reviewer #2: Yes

Reviewer #3: Yes

5. Review Comments to the Author

Reviewer #1: Summary

The manuscript with the title ‘concurrent of stunting and overweight/obesity among children’, is looking into the co-existence of stunting and overweight and associated factors within children (6-23 months) in Ethiopia. The data used is extracted from the EDHS 2016 and analyses over 2000 children using hierarchical linear regression models including three categories/factors (distal, intermediate and proximal). Prevalence of double burden of malnutrition was approx. 3%. Associated factors such as household wealth, birth size, gender, and intake of supplementation significantly associated with the occurrences of the stunting and overweight in children.

The intention and the idea of the paper are relevant and underline the current shift into nutrition transition in African countries, especially in rural areas. In addition, the findings demonstrate that policy makers need to rethink their agenda to include all forms of malnutrition in their interventions.

However, the data available could be explored further and the authors should revise the language and structure of the manuscript to improve readability. I really like the overall idea of the paper though paper fails to relate to previous research in the area, especially the link from the results to IYCF recommendations and policy recommendations and opportunities. Further, the paper could benefit to include other scenarios of double burden of malnutrition, such as wasting and overweight/obesity, micro nutrient deficiencies and overweight/obesity.

1. Major issues

Introduction

While the topic is very relevant, the authors work with different wording of co-existence (Dual vs. double burden from line 82 on wards). I would advise to revise the manuscript using one definition, which is in line with the WHO guidelines and the recent published Lancet series of double burden of malnutrition. In addition, the introduction should include the different possibilities of defining double burden of malnutrition and which indicators are used. To my understanding there is no clear definition on the indicators used to explore and define the double burden of malnutrition phenomenon.

The paper uses stunting and overweight as indicators of double burden of malnutrition in children, which is a valid definition. Thus, the paper would benefit from a more detailed literature review (line 101 ff) to justify their choice. The authors indicated several papers in the introduction, but more information of those papers are needed to understand the argumentation and literature gap. Especially, if there really is now other evidence from other Eastern African countries this should be worked out more clearly.

Methods

I advise the authors to consider restructuring the method section to be more informative and specific. The differentiation of the section study subjects, and study design and data are not entirely clear to me. The section would benefit from additional background on the secondary data sets (How often is the data collected, include the relevant type of data which is collected?).

To be more specific:

- Line 124-128, was this sampling structure applied to the data set? I assume it is the strategy for the EDHS sampling, but this is not coming out clearly.

- Line 128 reference missing

- How was dealt with outliers calculating anthropometric measurements? How were indicators calculated and which cut-off points used? How many cases needed to be dropped? (line 152)

- Dietary/non-dietary data? In Table 1 dietary data is listed, please explain how this data was collected.

- Why was BAZ and not WAZ used for the calculation of child overweight?

Data analysis section should be expanded and clarified to ensure that readers understand the applied methods and approach. A table of factors under consideration would be useful in the explanations of distal, intermediate and proximal factors.

The authors list the cut-off point of p<0.2 within the bivariate analyses, however not all factors below the cut-off point are used in the regression model (see table 1 and 2 in the results section). Where does the cut-off p<0.2 come from? This technical detail should be explained further to avoid confusion.

Results

The authors jump right into the presentation of the bivariate analyses. This section would benefit from a background characteristics/ descriptive statistics table to understand the study population, before the results of the bivariate analyses is presented

In line 184f: The authors state that table 1 will presents background characteristics, but the naming of table displays bivariate analyses, which is misleading

Table 2 includes variables which are above the cut-off point of below p<0.2, therefore details are missing as of why those variables were also used in the regression analyses.

Discussion

The authors should consider rewriting the discussion section to structure according to the presentation of the results and to avoid repetition of the findings. In addition, the authors should use the literature review to discuss current findings with results from other east African countries.

In line 276, the authors discuss the finding that children in rural households are more effected compared to urban household, which is a very interesting and relevant finding, this needs to be further discussed and explained on why it is so interesting. Is there other literature that could strengthen the results?

Likewise, in line 277 ff it is stated that boy are more affected compared to girls. However, the explanation and reason behind is not clear to me. Please revise and bring out the point more strongly.

IYCF guidelines are addressed within the abstract and the conclusion, but not in the discussion. I would advise to discuss the current findings in relation to the guidelines and policy recommendations.

Conclusion

While the authors discuss various details within the discussion section, a new point is brought up within the conclusion which has not been mentioned before. In my opinion the conclusion should be round up from the current results and the discussion and therefore, should lead to action points and suggestions. The conclusion should not explore new discussion points. Therefore, the sections need to be revised to bring out the key message and recommendation found and discussed.

1.2. Minor issues

Introduction

The authors mentioned within the abstract IYCF indicators, which are not mentioned within the introduction anymore. Those indicators could underline the importance of the analysis of the associated factors.

The introduction would benefit from more detailed description of the study area. Prevalence’s of overweight/ obese children (Line 79f) would increase visibility of problem and rise of nutrition transition and underline the purpose of the study.

Method

I would advise the author to rename the data analyses section into statistical method/design. As the section highlights the economic models/ approach used in the analyses.

Results

I suggest the authors recheck the results mentioned within text and displayed at tables to rule out any inconsistency.

Within the method section and the results section two different names are used for the categories/factors (basic vs. distal, underlying vs. intermediate, immediate vs. proximal) please be more consistent. It would be easier if the authors could explain the different categories and use a uniform definition.

The layout of Table 1 and 2 could be clearer, and additional explanations to different variables would underline the current results, such as the type of variables used (scores/ dummy variables etc.), to use a common war to display significant level and a detailed description of the abbreviations used within the table (Table 2 meaning of ref.?) In addition, i advise the authors to revise the results section to be sure that the results within the table are supporting the results within the text.

Any other:

In the abbreviation section, abbreviations are mentioned which are either not within the manuscript or the other way around, the authors should check the abbreviation and references section carefully to avoid any inconsistency and misspelling (Example: Reference 45).

Reviewer #2: This is a well-written manuscript that summarizes research that may be of interest and value for the professional knowledge based related to cross-national research on concurrent of stunting and overweight/obesity among children.

Reviewer #3: Concurrent of stunting and overweight/obesity among children:evidence from Ethiopia

Reviewer comments

General

Investigation of the combination of stunting and overweight/obesity and predictors/determinants is an important topic. However, this paper needs major work on the methods and I suggest very strongly that the determinants of stunting and overweight/obesity in this group is investigated in a similar manner as was done for the combination and then integrated into the discussion to add more body and value to this paper in terms of recommendations for prevention/management of malnutrition (double burden at population, household and individual levels).

Although the language is reasonable to good, the entire paper needs a further level of language editing to ensure that it is correct.

Abstract

Not commented on yet pending suggested additions.

Methods

Please note that the flow and detail included in the methods sections need serious attention.

Please start this section with the ‘Study design and data source section’ (currently line 29) that should cover the following:

• Design: investigation of double burden of malnutrition defined as co-existence of stunting and overweight/obesity within the same child and associated factors using data from the 2015 EDHS (what does this stand for?) The associated factors need to be mentioned here i.e the concepts of distal, intermediate and proximal factors and what they entail need to be explained here.

• More detail on this national survey than is currently included in this section (some of which is e.g. presented in lines 141-145)

• The sampling design and procedure applied in the national survey (it is not sufficient to indicate that detail was published elsewhere). The representativeness of the final sample of Ethiopian children in the target age group should also be alluded to.

• Ethics approval for the overarching national study and then this secondary data analysis can be covered here.

Then the ‘Study subjects’ section follows that covers the criteria stated for inclusion in the current analysis and the final number included.

The next heading should be ‘Anthropometric measures and interpretation’ and should firstly provide detail on how weight and length were measured (more detail than given), followed by interpretation criteria. The model and manufacturers of equipment for measurement of height and weight also need to be included. Please also include the definition (cut-offs) of combined stunting-overweight/obesity as HAZ<-2SD and BAZ>2SD (I assume the abbreviation, CSO, used in the statistics section refers to this); although it may seem self-explanatory. Please provide a reference for the WHO standards. The next heading should be ‘Assessment of associated factors’. Were these factors associated factors determined using a questionnaire, if yes, was it interviewer administered, was the mother/primary care giver the interviewee? What was covered in the questionnaire (provide indication of the different sections – including questions, instruments used), how was the questionnaire developed, pilot tested etc.? In my view the questions that covered distal and intermediate factors were very limited, and may thus not have been sensitive enough to reflect these two levels of factors. This should be discussed in the limitations sections.

The next heading should be ‘Data collection procedures’ which should cover where and when data was collected by whom (= fieldworkers and their training)

Next heading would be ‘Statistical analysis’. Please clarify/address the following:

• Please indicate that CSO (YES/NO) was the dependant variable in each of the three regression models.

• Why was p<0.2 used as an indication for inclusion in regression models

• Lines 165-169: please remove: duplication of previous information

• Lines 176-181: This explanation is not clear – please improve.

• Was co-dependence between variables considered in the regression analyses?

Results

Table 1: Only a few of the variables included in table 1 are interpreted in the text (repetition of exact results in the text – e.g. % 12 months and 5 over 12 months is not necessary if all the figures are presented in the table), only main trends for the different variables need to be included in the text). Please include a more comprehensive interpretation of the table in the text starting from proximal, then intermediate and then last the distal effects (this is also the order in which the variables were included in the regressions models). Table 1 format: Please correct the first variable so that data for ‘urban’ is aligned with the variable name (place of residence); in the foot note indicate that the chi square test was ‘Pearson’s’

Table 2: Please indicate in the text when introducing Table 2 that it covered all three models (model 1= distal factors only, model 2=distal + proximal factors, model 3=distal, underlying + proximal factors). Lines 213-224: it is not necessary to repeat all the results included in the table in the text. The interpretation of the results in the table can be written as follows in the text:

Determinants of CSO included lowest wealth index category (AOR=2.07), being a male child (AOR=1.6), being older than 12 (AOR=1.76), having been small at birth (AOR=2.53) not having taken vitamin A supplement within the previous six months (AOR=1.91)

Table 2 format: Please explain what each model involved in the Footnote to the table. Suggest to revise the table title as follows:

Table 2: Hierarchical multiple logistic regression analysis to determine basic, underlying and proximal determinants of CSO

Discussion

No detailed comments pending the decision to include investigation of determinants of stunting and overweight/obesity as well as suggested.

Please take care not to interpret and discuss non-significant results in such a manner that it seems to be deemed a determinant: example: Line 272-276 ‘Children who reside in the rural areas had a higher risk of being concurrently stunting and overweight though insignificant……’

6. PLOS authors have the option to publish the peer review history of their article (what does this mean?). If published, this will include your full peer review and any attached files.

Reviewer #1: No

Reviewer #2: No

Reviewer #3: No

---

## [Author Response · Author response to Decision Letter 0]

22 May 2020

Reviewer comments

Title: Concurrent of stunting and overweight/obesity among children: evidence from Ethiopia

Date: 22/05/2020

Dear Editor,

We would like to thank you and the reviewers for the comments provided to our manuscript which help to improve the quality of our manuscript. Our response is as follows:

Please note we have made changes in our study population and analyzed all children <5 years of age and not 6-23 months. Thus, dietary factors were not considered in the new analysis since they are only applicable for children <2 years. 

Reviewers Comments Author’s Revision and Response

Reviewer 1 

The intention and the idea of the paper are relevant and underline the current shift into nutrition transition in African countries, especially in rural areas. In addition, the findings demonstrate that policy makers need to rethink their agenda to include all forms of malnutrition in their interventions. We would like to thank the reviewer for the positive appraisal of the current work.

However, the data available could be explored further and the authors should revise the language and structure of the manuscript to improve readability. Agreed, the language revised 

I really like the overall idea of the paper though paper fails to relate to previous research in the area, especially the link from the results to IYCF recommendations and policy recommendations and opportunities. Agreed, as per the reviewer recommendation additional papers were reviewed and included discussion section. See line 566-578. See an excerpt from the manuscript:

“A recent analysis conducted by WHO has also showed that policies that address undernutrition often times do not include overweight/obesity and the vice versa. Thus, integrated interventions, programs and policies that have the potential to improve the nutrition outcomes across the spectrum of malnutrition is of paramount importance to simultaneously reduce the risk or burden of both undernutrition and overweight/obesity”

Further, the paper could benefit to include other scenarios of double burden of malnutrition, such as wasting and overweight/obesity, micro nutrient deficiencies and overweight/obesity. We would like to thank the reviewer for the suggestion but we feel those could be research topics by themselves and we are currently working on those topics as separate research topics particularly on overweight/obesity and micronutrient deficiencies. Besides, it is beyond the objective of the current work.

While the topic is very relevant, the authors work with different wording of co-existence (Dual vs. double burden from line 82 onwards). I would advise to revise the manuscript using one definition, which is in line with the WHO guidelines and the recent published Lancet series of double burden of malnutrition. Agreed, the manuscript revised and used ‘double burden of malnutrition’ consistently throughout the document.

The introduction should include the different possibilities of defining double burden of malnutrition and which indicators are used. Agreed, we mentioned the fact the there are no uniform definition of double burden indicators. See lines 111-119. See an excerpt from the manuscript:

“…there is no uniform definition of double burden indicators. Indicators of child malnutrition used by the reviewed literatures are combination of Height for age-z score (HAZ) and micronutrient deficiency, BMI for age-z-score and HAZ, Weight for height z-score (WHZ) and HAZ and Weight for age z-score (WAZ) and HAZ. These differences in measurement, make comparison among studies difficult”

The paper uses stunting and overweight as indicators of double burden of malnutrition in children, which is a valid definition. Thus, the paper would benefit from a more detailed literature review (line 101 ff) to justify their choice Agreed, more paper reviewed and included papers on stunting as well as overweight/obesity. See line 73-83.

The authors indicated several papers in the introduction, but more information of those papers is needed to understand the argumentation and literature gap. Especially, if there really is now other evidence from other Eastern African countries this should be worked out more clearly. Agreed, more papers from the Eastern African were included and many of these studies analyzed double burden within households by looking at pairs of overweight mothers and undernourished children. They were more of prevalence and trends of dual burden and hardly examined factors associated with the double burden. See line 134-137.

I advise the authors to consider restructuring the method section to be more informative and specific. The differentiation of the section study subjects, and study design and data are not entirely clear to me. The section would benefit from additional background on the secondary data sets (How often is the data collected, include the relevant type of data which is collected?). Agreed, the method section restricted and started with study design and data source section which is more detailed, followed by subject, then anthropometric measures and interpretation, data collection procedure and statistical analysis. See lines 157-269.

Line 124-128, was this sampling structure applied to the data set? I assume it is the strategy for the EDHS sampling, but this is not coming out clearly. No, this was strategy for the EDHS procedure, detailed sample design and procedure of EDHS included. See lines 158-181.

Line 128 reference missing 

How was dealt with outliers calculating anthropometric measurements? How were indicators calculated and which cut-off points used? How many cases needed to be dropped? Flagged values were dropped and children’s weight and length/height measurements were transformed into sex- and age-specific Z-scores using WHO 2006 growth standards see lines 202 to 204. 122 cases were dropped due to flagged values. see line 183 to 89. Figure also included to show the flow of sample selection. See Figure 1

Dietary/non-dietary data? In Table 1 dietary data is listed, please explain how this data was collected Agreed, we tried to explain the data collection procedure. See data collection procedure section (lines 211-226). 

Why was BAZ and not WAZ used for the calculation of child overweight? We used BAZ as alternative of WHZ because both indicators demonstrated high agreement with low misclassification (according to Kayla., et.al,2016). WAZ is composite indicator and not commonly used to calculate child overweight and instead WHZ or BAZ used. We found an overall prevalence of overweight/obesity was higher compared to estimates based on weight for height z-score. 3.86% and 3.5% for BAZ and WHZ respectively.

Data analysis section should be expanded and clarified to ensure that readers understand the applied methods and approach Agreed, this section expanded. See lines 227-269

A table of factors under consideration would be useful in the explanations of distal, intermediate and proximal factors Agreed, additional table added to explain the different variables in the manuscript. See table 1.

The authors list the cut-off point of p<0.2 within the bivariate analyses, however not all factors below the cut-off point are used in the regression model (see table 1 and 2 in the results section). Where does the cut-off p<0.2 come from? This technical detail should be explained further to avoid confusion. The p< 0.2 was cut off point for bivariate regression analysis and not the chi-square. One column for crude odds ratio (COR) added in each model to clear the confusion. Only variables that demonstrated P<0.2 were included in hierarchical regression models. This is process is called ‘purposeful selection of covariates’ and suggested by previous studies (cited in the manuscript, see line 242-248 and table 2). We also kept some important variables regardless of their significance. See the excerpt from the manuscript:

“Regardless of significance, basic factors, as the primary independent variables, were retained in the final regression model. Since age and sex are important factors for outcomes in under five children, they were also retained in the final models”

The authors jump right into the presentation of the bivariate analyses. This section would benefit from a background characteristics/ descriptive statistics table to understand the study population, before the results of the bivariate analyses is presented. Thank you for the suggestion, we felt the table can serve both purposes by describing background characteristics and the same time determine differences in proportions.

In line 184f: The authors state that table 1 will presents background characteristics, but the naming of table displays bivariate analyses, which is misleading Agreed, the statement paraphrased as “The background characteristics of children and prevalence of CSO across different covariates are presented in Table 2”. The table displays bivariate analysis in a sense it shows the differences in the proportions across different child characteristics.

Table 2 includes variables which are above the cut-off point of below p<0.2, therefore details are missing as of why those variables were also used in the regression analyses. Agreed, it was misleading. P <0.2 was cut off point for bivariate regression analysis and not the chi-square. One column for crude odds ratio (COR) added in each model to clear the confusion. Only variables that demonstrated P<0.1 were included in hierarchical regression models. See table 3,4 &5

The authors should consider rewriting the discussion section to structure according to the presentation of the results and to avoid repetition of the findings. In addition, the authors should use the literature review to discuss current findings with results from other east African countries. Agreed. Literatures from East African countries discussed in this section. It is also structured according to the presentation of the result. See lines 442-444

In line 276, the authors discuss the finding that children in rural households are more effected compared to urban household, which is a very interesting and relevant finding, this needs to be further discussed and explained on why it is so interesting. Is there other literature that could strengthen the results? Removed, the 3rd reviewer suggested not discuss non-significant factors. But we find association between stunting and place of residence and was further discussed. See lines 497-501.

Likewise, in line 277 ff it is stated that boy are more affected compared to girls. However, the explanation and reason behind is not clear to me. Please revise and bring out the point more strongly. Agreed, the reasons explained. See lines 526-533.

IYCF guidelines are addressed within the abstract and the conclusion, but not in the discussion. I would advise to discuss the current findings in relation to the guidelines and policy recommendations. Agreed, findings discussed in relation to current interventions and policies. See lines 572-578.

 While the authors discuss various details within the discussion section, a new point is brought up within the conclusion which has not been mentioned before. In my opinion the conclusion should be round up from the current results and the discussion and therefore, should lead to action points and suggestions. The conclusion should not explore new discussion points. Therefore, the sections need to be revised to bring out the key message and recommendation found and discussed. Agreed, the conclusion revised as follows:

“In conclusion, our study provided evidence on the co-existence of stunting and overweight/obesity among infants and young children in Ethiopia. CSO was associated with various factors originating from community and child levels. Therefore, identifying children at risk of growth flattering and excess weight gain provides nutrition policies and programs in Ethiopia and beyond with an opportunity of earlier interventions through improving sanitation, dietary quality by targeting children under 12 years of age and those living in Agrarian regions of Ethiopia”

The authors mentioned within the abstract IYCF indicators, which are not mentioned within the introduction anymore. Those indicators could underline the importance of the analysis of the associated factors. Agreed, but it is not any more applicable in the current analysis since we have used data for children aged 0-59 months and not 6-23 months.

The introduction would benefit from more detailed description of the study area. Prevalence’s of overweight/ obese children (Line 79f) would increase visibility of problem and rise of nutrition transition and underline the purpose of the study. Agreed, description on the study area added. See lines 89-95.

I would advise the author to rename the data analyses section into statistical method/design. As the section highlights the economic models/ approach used in the analyses. Agreed, the section renamed as ‘statistical analysis’

I suggest the authors recheck the results mentioned within text and displayed at tables to rule out any inconsistency. Agreed, the result in the table and text crosschecked no inconsistency found.

Within the method section and the results section two different names are used for the categories/factors (basic vs. distal, underlying vs. intermediate, immediate vs. proximal) please be more consistent. It would be easier if the authors could explain the different categories and use a uniform definition. Agreed, the different categories explained and uniform definition used. We used basic, underlying and immediate factors consistently throughout the manuscript. 

The layout of Table 1 and 2 could be clearer, and additional explanations to different variables would underline the current results, such as the type of variables used (scores/ dummy variables etc.), to use a common war to display significant level and a detailed description of the abbreviations used within the table (Table 2 meaning of ref.?) Agreed, additional table included to explain different variables. See table 1.

In addition, I advise the authors to revise the results section to be sure that the results within the table are supporting the results within the text. Agreed, revised accordingly and found no inconsistencies.

In the abbreviation section, abbreviations are mentioned which are either not within the manuscript or the other way around, the authors should check the abbreviation and references section carefully to avoid any inconsistency and misspelling (Example: Reference 45) Agreed, inconsistencies check and corrected accordingly 

Reviewer 2 

This is a well-written manuscript that summarizes research that may be of interest and value for the professional knowledge based related to cross-national research on concurrent of stunting and overweight/obesity among children. We would to thank the reviewer for the positive appraisal of the current work.

Reviewer 3 

Investigation of the combination of stunting and overweight/obesity and predictors/determinants is an important topic. We would to thank the reviewer for the positive appraisal of the current work.

However, this paper needs major work on the methods and I suggest very strongly that the determinants of stunting and overweight/obesity in this group is investigated in a similar manner as was done for the combination and then integrated into the discussion to add more body and value to this paper in terms of recommendations for prevention/management of malnutrition (double burden at population, household and individual levels). Agreed, method part improved and determinants of stunting and overweight investigated and integrated into the discussion section.

Although the language is reasonable to good, the entire paper needs a further level of language editing to ensure that it is correct. Agreed, the language edited improved and we hope it has improved in this version.

Please start this section with the ‘Study design and data source section’ (currently line 29) that should cover the following:

• Design: investigation of double burden of malnutrition defined as co-existence of stunting and overweight/obesity within the same child and associated factors using data from the 2015 EDHS (what does this stand for?) The associated factors need to be mentioned here i.e the concepts of distal, intermediate and proximal factors and what they entail need to be explained here

• More detail on this national survey than is currently included in this section (some of which is e.g. presented in lines 141-145)

• The sampling design and procedure applied in the national survey (it is not sufficient to indicate that detail was published elsewhere). The representativeness of the final sample of Ethiopian children in the target age group should also be alluded to.

• Ethics approval for the overarching national study and then this secondary data analysis can be covered here. 

• Agreed, revised accordingly and the concept of basic, underlying and immediate factors explained. See lines 172-177.

• Agreed, detailed sample design and procedure of EDHS included. See lines 158-181.

• Ethics of approval covered under separate sub-heading. See lines 270-276.

Then the ‘Study subjects’ section follows that covers the criteria stated for inclusion in the current analysis and the final number included. Agreed and revised accordingly. See lines 182-189.

The next heading should be ‘Anthropometric measures and interpretation’ and should firstly provide detail on how weight and length were measured (more detail than given), followed by interpretation criteria. The model and manufacturers of equipment for measurement of height and weight also need to be included. Please also include the definition (cut-offs) of combined stunting-overweight/obesity as HAZ<-2SD and BAZ>2SD (I assume the abbreviation, CSO, used in the statistics section refers to this); although it may seem self-explanatory. Please provide a reference for the WHO standards. Agreed, the section revised accordingly. Measurements were described in detail and cut off of CSO stated and reference provided. See 191-204.

The next heading should be ‘Assessment of associated factors. Were these factors associated factors determined using a questionnaire, if yes, it was interviewer administered, was the mother/primary care giver the interviewee? What was covered in the questionnaire (provide indication of the different sections – including questions, instruments used), how was the questionnaire developed, pilot tested etc.? In my view the questions that covered distal and intermediate factors were very limited, and may thus not have been sensitive enough to reflect these two levels of factors. This should be discussed in the limitation’s sections. Agreed, instrument used, piloting, etc. covered in data collection procedure section. brief explanation of the factors and how was data collected explained under assessment of associated factors and data collection procedure. A detailed description these factors is presented in Table 1. 

The next heading should be ‘Data collection procedures’ which should cover where and when data was collected by whom (= fieldworkers and their training) Agreed, revised accordingly. See lines 211-226.

Next heading would be ‘Statistical analysis’. Please clarify/address the following:

• Please indicate that CSO (YES/NO) was the dependent variable in each of the three regression models.

• Why was p<0.2 used as an indication for inclusion in regression models

• Lines 165-169: please remove: duplication of previous information

• Lines 176-181: This explanation is not clear – please improve.

• Was co-dependence between variables considered in the regression analyses? Agreed and renamed the heading ‘statistical analysis’

• Agreed, CSO indicated as dependent variable

• Agreed, the p< 0.2 was cut off point for bivariate regression analysis and not the chi-square. One column for crude odds ratio (COR) added in each model to clear the confusion. Only variables that demonstrated P<0.2 were included in hierarchical regression models. This is process is called ‘purposeful selection of covariates’ and suggested by previous studies

• Agreed, statement added to make it clear. See lines 266-268. 

• We used VIF command to test for multicollinearity and the mean VIF was 1.20 and all variables had VIF less than 2.0. 

Table 1: Only a few of the variables included in table 1 are interpreted in the text (repetition of exact results in the text – e.g. % 12 months and 5 over 12 months is not necessary if all the figures are presented in the table), only main trends for the different variables need to be included in the text). Please include a more comprehensive interpretation of the table in the text starting from proximal, then intermediate and then last the distal effects (this is also the order in which the variables were included in the regressions models). Agreed, more comprehensive interpretation of the table included in the text starting from basic, underlying and then immediate factors. See lines 

Table 1 format: Please correct the first variable so that data for ‘urban’ is aligned with the variable name (place of residence); in the foot note indicate that the chi square test was ‘Pearson’s’ Agreed and corrected accordingly 

Table 2: Please indicate in the text when introducing Table 2 that it covered all three models (model 1= distal factors only, model 2=distal + proximal factors, model 3=distal, underlying + proximal factors). Agreed and indicated in the text that table 3 covers all the three models.

Lines 213-224: it is not necessary to repeat all the results included in the table in the text. The interpretation of the results in the table can be written as follows in the text:

Determinants of CSO included lowest wealth index category (AOR=2.07), being a male child (AOR=1.6), being older than 12 (AOR=1.76), having been small at birth (AOR=2.53) not having taken vitamin A supplement within the previous six months (AOR=1.91).

 Agreed and revised as per the reviewer suggestion.

Table 2 format: Please explain what each model involved in the Footnote to the table. 

 Agreed and explained what each model involved in the footnote of the table 3,4&5.

Suggest to revise the table title as follows:

Table 2: Hierarchical multiple logistic regression analysis to determine basic, underlying and proximal determinants of CSO Agreed and revised the table title as per the reviewer suggestion 

Please take care not to interpret and discuss non-significant results in such a manner that it seems to be deemed a determinant: example: Line 272-276 ‘Children who reside in the rural areas had a higher risk of being concurrently stunting and overweight though insignificant……’ Agreed, non-significant result not discussed in the discussion section.

---

## [Decision Letter · Decision Letter 1]

3 Aug 2020

PONE-D-20-03785R1

Concurrent of stunting and overweight/obesity among children:evidence from Ethiopia

PLOS ONE

Dear Dr. Farah,

Thank you for submitting your manuscript to PLOS ONE. After careful consideration, we feel that it has merit but does not fully meet PLOS ONE’s publication criteria as it currently stands. Therefore, we invite you to submit a revised version of the manuscript that addresses the points raised during the review process.

We look forward to receiving your revised manuscript.

Kind regards,

Nili Steinberg

Academic Editor

PLOS ONE

Reviewers' comments:

Reviewer's Responses to Questions

**Comments to the Author**

1. If the authors have adequately addressed your comments raised in a previous round of review and you feel that this manuscript is now acceptable for publication, you may indicate that here to bypass the “Comments to the Author” section, enter your conflict of interest statement in the “Confidential to Editor” section, and submit your "Accept" recommendation.

Reviewer #1: (No Response)

Reviewer #2: All comments have been addressed

2. Is the manuscript technically sound, and do the data support the conclusions?

Reviewer #1: Yes

Reviewer #2: Yes

3. Has the statistical analysis been performed appropriately and rigorously? 

Reviewer #1: Yes

Reviewer #2: Yes

4. Have the authors made all data underlying the findings in their manuscript fully available?

Reviewer #1: Yes

Reviewer #2: Yes

5. Is the manuscript presented in an intelligible fashion and written in standard English?

Reviewer #1: No

Reviewer #2: Yes

6. Review Comments to the Author

Reviewer #1: Thanks to the authors for the extensive revision of the manuscript, while most of the manuscript has improved, I remain with further comments to the authors for clarification of the manuscript. Additionally, I believe the manuscript could further benefit from language edition to improve readability. First, I want to mention that my line references are to the manuscript without track changes (the first one within the PDF).

Thank you for mentioning upfront the shift of focus of the age bracket of the children from 6-23 towards 6-59 months of age. However, could the authors please elaborate further on why the shift was done. The authors mentioned dietary data availability as one point to expand the age bracket. Within the discussion dietary effects are used to explain the difference between agrarian and pastoralists. The available dietary data for children 6-23 month, if analysed could strengthen this point (Line 476 ff) as a consistency check. I believe these results were mentioned in the previous version of the manuscript and could be included as a footnote for the sub-sample.

While the manuscript improved substantially, I still would advise to revision the manuscript further for readability and uniformity. Different writing styles are used within the manuscript (cooccurrence vs co-occurrence (line 77 vs. 108), presentation of results numeric and verbal (line 84 vs 92). Additionally, have a keen eye on the style of literature which is inconsistent. While the authors tried to in cooperate the comment on the naming of the double burden of malnutrition please re-check for consistency.

Within the Abstract the authors mention in line 55 ‘younger than 12 months’. Within the conclusion ‘younger than 12 years of age’ is mentioned (line 601 f). Please revise.

I appreciate the effort and work the authors put into rewriting the method section as requested by the reviewers, while the new structure is more intuitive, some information is in my eyes still explained within the wrong section. The first section study design and data source still fail to name the data set, the name (EDHS line 188) are introduced within the next section study subjects. I, therefore, advise revising the section making sure the present information is fitting to the current section naming.

Thanks to the authors to include further information on the selection of the children. I still would like to get more clarification on the sample size construction. The authors mentioned children not alive (lien 184 ff and figure 1) were dropped for the analysis. Please elaborate on how this data was collected or even present within the data set.

While the authors took up the suggestion to remain with one name of the categorization of basic, underlying and immediate factors, within the manuscript the previous names are still present. Please revised to be consistent throughout the manuscript (see as example line 331).

Line 251ff sentence is repetitive please review.

Thanks for the inclusion of Table 1, which displays information about the associated factors. However, the Table would benefit from revision be improved readability and visibility for the reader. Please remove all the information not necessary to the construction of the variable. For example, CSO only needs the definition applied to the dummy variable. Please revise the Table numbering and layout. Most tables have several table numbers listed.

Sentence in line 432 is not clear to me, please revise.

Reviewer #2: (No Response)

7. PLOS authors have the option to publish the peer review history of their article (what does this mean?). If published, this will include your full peer review and any attached files.

Reviewer #1: No

Reviewer #2: No

---

## [Author Response · Author response to Decision Letter 1]

17 Sep 2020

Dear Editor,

We would like to thank you and the reviewers for the comments provided to our manuscript which help to improve the quality of our manuscript. Our response is as follows:

Reviewer 1 

Thanks to the authors for the extensive revision of the manuscript, while most of the manuscript has improved, I remain with further comments to the authors for clarification of the manuscript. Additionally, I believe the manuscript could further benefit from language edition to improve readability. First, I want to mention that my line references are to the manuscript without track changes (the first one within the PDF). 

-We would like to thank reviewer for the positive appraisal on the revised work. 

-Language improved. Dr. Olusola Oladeji has edited the manuscript. 

Thank you for mentioning upfront the shift of focus of the age bracket of the children from 6-23 towards 6-59 months of age. However, could the authors please elaborate further on why the shift was done. 

-The shift of focus from age group of 6-23 to 6-59 months was mainly for generalizability. Besides, most of the literatures cited in our work focused on 6-59 months. 

The authors mentioned dietary data availability as one point to expand the age bracket. Within the discussion dietary effects are used to explain the difference between agrarian and pastoralists. The available dietary data for children 6-23 month, if analyzed could strengthen this point (Line 476 ff) as a consistency check. I believe these results were mentioned in the previous version of the manuscript and could be included as a footnote for the sub-sample. 

-Agreed, and we would like to thank the reviewer for this valuable comment. Sub sample analysis done for data of children aged 6-23 months and found that 62% and only 15% of children from pastoralist and agrarian communities respectively consumed dairy products (P<0.001) *

-* Pearson chi-squared test 

While the manuscript improved substantially, I still would advise to revision the manuscript further for readability and uniformity. Different writing styles are used within the manuscript (cooccurrence vs co-occurrence (line 77 vs. 108), presentation of results numeric and verbal (line 84 vs 92). Additionally, have a keen eye on the style of literature which is inconsistent. While the authors tried to in cooperate the comment on the naming of the double burden of malnutrition please re-check for consistency 

-Agreed, consistency rechecked and “concurrence” used throughout the manuscript. 

-Agreed, numeric used consistently throughout the manuscript

-Consistency of literature rechecked 

Within the Abstract the authors mention in line 55 ‘younger than 12 months. Within the conclusion ‘younger than 12 years of age’ is mentioned (line 601 f). Please revise. 

-Agreed, revised and replaced with younger than 5 years.

I appreciate the effort and work the authors put into rewriting the method section as requested by the reviewers, while the new structure is more intuitive, some information is in my eyes still explained within the wrong section. The first section study design and data source still fail to name the data set, the name (EDHS line 188) are introduced within the next section study subjects. I, therefore, advise revising the section making sure the present information is fitting to the current section naming. 

-Agreed, section revised 

Thanks to the authors to include further information on the selection of the children. I still would like to get more clarification on the sample size construction. The authors mentioned children not alive (lien 184 ff and figure 1) were dropped for the analysis. Please elaborate on how this data was collected or even present within the data set. 

-DHS surveys use a full birth history which is a complete list of all children a mother has ever given birth to including their date of birth, sex, survival status, age (if alive), and age at death (if died). This is the form of birth history found in the majority of DHS surveys.

While the authors took up the suggestion to remain with one name of the categorization of basic, underlying and immediate factors, within the manuscript the previous names are still present. Please revised to be consistent throughout the manuscript (see as example line 331).

Line 251ff sentence is repetitive please review. 

-Agreed, basic, underlying and immediate factors were used consistently throughout the manuscript.

Thanks for the inclusion of Table 1, which displays information about the associated factors. However, the Table would benefit from revision be improved readability and visibility for the reader. Please remove all the information not necessary to the construction of the variable. For example, CSO only needs the definition applied to the dummy variable. Please revise the Table numbering and layout. Most tables have several table numbers listed. 

-Agreed, unnecessary information to construct the variables removed and table layout and numbering revised.

Sentence in line 432 is not clear to me, please revise. 

-Agreed and revised as follows: “The immediate factors found associated with higher odds of CSO were age younger than 12 months, no history of infection and not having received deworming for the last six months”.

---

## [Decision Letter · Decision Letter 2]

2 Nov 2020

PONE-D-20-03785R2

Concurrence of stunting and overweight/obesity among children:evidence from Ethiopia

PLOS ONE

Dear Dr. Farah,

Thank you for submitting your manuscript to PLOS ONE. After careful consideration, we feel that it has merit but does not fully meet PLOS ONE’s publication criteria as it currently stands. Therefore, we invite you to submit a revised version of the manuscript that addresses the points raised during the review process.

We look forward to receiving your revised manuscript.

Kind regards,

Nili Steinberg

Academic Editor

PLOS ONE

Reviewers' comments:

Reviewer's Responses to Questions

**Comments to the Author**

1. If the authors have adequately addressed your comments raised in a previous round of review and you feel that this manuscript is now acceptable for publication, you may indicate that here to bypass the “Comments to the Author” section, enter your conflict of interest statement in the “Confidential to Editor” section, and submit your "Accept" recommendation.

Reviewer #1: (No Response)

2. Is the manuscript technically sound, and do the data support the conclusions?

Reviewer #1: Yes

3. Has the statistical analysis been performed appropriately and rigorously? 

Reviewer #1: Yes

4. Have the authors made all data underlying the findings in their manuscript fully available?

Reviewer #1: Yes

5. Is the manuscript presented in an intelligible fashion and written in standard English?

Reviewer #1: Yes

6. Review Comments to the Author

Reviewer #1: Thanks to the authors for submitting the revised manuscript. I remain with some minor suggestions and comments. The line references to these suggestions and comments are based on the manuscript with track changes (second version in the PDF file).

The manuscript has improved language-wise, still please have a keen eye on the writing style age of the children (for example line 83 vs. line 109 or line 134 or line 615f). Please revise the following sentences as they hold repetitive information: line 87f.

Please revise the sentence in line 137ff. which is unclear to me.

Thanks for the revision at the study subject area, I appreciate the idea of a figure, though I was not able to allocate the figure for the review. Please make sure to include the figure in the attachments.

Please check line 343 the table has two numbers.

7. PLOS authors have the option to publish the peer review history of their article (what does this mean?). If published, this will include your full peer review and any attached files.

Reviewer #1: No

---

## [Author Response · Author response to Decision Letter 2]

26 Nov 2020

The manuscript has improved language-wise, still please have a keen eye on the writing style age of the children (for example line 83 vs. line 109 or line 134 or line 615f). 

-We would like to thank the reviewer for point this out. “children under five years of age” was consistently used throughout the manuscript. 

Please revise the following sentences as they hold repetitive information: line 87f. Revised and the repetitions removed 

Please revise the sentence in line 137ff. which is unclear to me. 

-Revised as follows: “Children who are concurrently stunted and overweight/obese can be at greater risk of unhealthy development than normal children”

Thanks for the revision at the study subject area, I appreciate the idea of a figure, though I was not able to allocate the figure for the review. Please make sure to include the figure in the attachments. 

-Figure attached 

Please check line 343 the table has two numbers. 

-Repetition removed

---

## [Decision Letter · Decision Letter 3]

4 Jan 2021

Concurrence of stunting and overweight/obesity among children:evidence from Ethiopia

PONE-D-20-03785R3

Dear Dr. Farah,

We’re pleased to inform you that your manuscript has been judged scientifically suitable for publication and will be formally accepted for publication once it meets all outstanding technical requirements.

Kind regards,

Nili Steinberg

Academic Editor

PLOS ONE

Additional Editor Comments (optional):

Reviewers' comments:

Reviewer's Responses to Questions

**Comments to the Author**

1. If the authors have adequately addressed your comments raised in a previous round of review and you feel that this manuscript is now acceptable for publication, you may indicate that here to bypass the “Comments to the Author” section, enter your conflict of interest statement in the “Confidential to Editor” section, and submit your "Accept" recommendation.

Reviewer #1: All comments have been addressed

2. Is the manuscript technically sound, and do the data support the conclusions?

Reviewer #1: Yes

3. Has the statistical analysis been performed appropriately and rigorously? 

Reviewer #1: Yes

4. Have the authors made all data underlying the findings in their manuscript fully available?

Reviewer #1: Yes

5. Is the manuscript presented in an intelligible fashion and written in standard English?

Reviewer #1: Yes

6. Review Comments to the Author

Reviewer #1: Dear Authors,

Thank you very much for submitting the revised versions of your manuscript and the good work you put in. I remain with one comment, please revise and add a description to the Figure caption. So that the reader can understand the meaning and the attention of the figure without so that it can be a standalone figure, without searching in the manuscript for the explanation.

Many thanks.

7. PLOS authors have the option to publish the peer review history of their article (what does this mean?). If published, this will include your full peer review and any attached files.

Reviewer #1: No

---

## [Editor Report · Acceptance letter]

6 Jan 2021

PONE-D-20-03785R3 

Concurrence of stunting and overweight/obesity among children: evidence from Ethiopia 

Dear Dr. Farah:

I'm pleased to inform you that your manuscript has been deemed suitable for publication in PLOS ONE. Congratulations! Your manuscript is now with our production department. 

Kind regards, 

on behalf of

Dr. Nili Steinberg 

Academic Editor

PLOS ONE